# Separation of Boron from Geothermal Waters with Membrane System

**DOI:** 10.3390/membranes11040291

**Published:** 2021-04-16

**Authors:** Kadir Seval, Canan Onac, Ahmet Kaya, Abdullah Akdogan

**Affiliations:** 1Department of Chemical Engineering, Pamukkale University, Denizli 20070, Turkey; kadirseval@gmail.com (K.S.); akdogan@pau.edu.tr (A.A.); 2Department of Chemistry, Pamukkale University, Denizli 20070, Turkey; ahmetk@pau.edu.tr; 3Advanced Technology Application and Research Center, Pamukkale University, Denizli 20070, Turkey

**Keywords:** boron, polymeric membrane system, transport mechanism, geothermal water, separation

## Abstract

This study presents the separation and recovery of boron from geothermal waters with a polymeric membrane system and suggests a transport mechanism. The optimum relative parameters of the transport were examined. The recovery value of boron was 60.46% by using polymeric membrane system from prepared aquatic solution to the acceptor phase. The membrane’s capacity and selectivity of the transport process were examined. Kinetics values were calculated for each transport parameter. The optimum kinetic values were 1.4785 × 10^−6^ (s^−1^), 7.3273 × 10^−8^ (m/s), 13.5691 × 10^−8^ (mol/m^2^.s), 5.8174 × 10^−12^ (m^2^/s) for constant rate, permeability coefficient, flux, and diffusion coefficient, respectively. Boron was transported selectively and successfully from geothermal waters in the presence of other metal cations with 59.85% recovery value. This study indicates the application of real samples in polymeric membrane systems, which are very practical, economic, and easy to use for large-scale applications. The chemical and physical properties of polymer inclusion membranes (PIMs) offer the opportunity to be specially designed for specific applications.

## 1. Introduction

Boron is necessary for certain metabolic activities in irrigation water. The concentration of boron in irrigation water plays a critical role in the cultivation of many crops because it is one of the basic micronutrients. High boron concentrations cause toxic symptoms to appear in plants, decrease photosynthesis capacity and productivity of plants, affect plant growth negatively, and accelerate plant mortality in irrigation water [1,2]. Boron is also an important element in the feeding of humans and animals. The amount of boron that should be taken daily for an adult person is about 1 mg; this is achieved through normal food consumption. Therefore, there is no need for a complementary source of this element. Boron can be found naturally in ground water as industrial pollutants in surface waters or as a product of agricultural surface flows and decomposition plant materials. Boron is the most polluting toxic element among the other toxic elements in irrigation water of Turkey. Boron can be found naturally in ground water, surface waters as industrial pollutants, or as a product of agricultural surface flows and decomposition plant materials. The concentration of boron compounds increases in wastewaters, which finds common use in many industrial applications. Boron is found in domestic wastewater as a result of being added to detergents and cleaning products; various chemicals that are used in agriculture products are the main sources of boron in wastewater. 

There are many various separation and purification methods for the removal of boron from drinking and irrigation waters, which adversely affect agricultural production and human health [3,4,5,6]. Due to the rapid decline of freshwater resources in the world, the use of sea water containing approximately 5 mg/L boron as an alternative water source for both drinking water and irrigation water has become attractive [7]. Therefore, the control of the concentration of boron in treated seawater is very important for different applications. 

Many useful methods are used for boron removal. The most used of these methods (in order) are: adsorption, chemical precipitation, ion exchange, reverse osmosis (RO), electro-dialysis (ED), electro-coagulation, Donnan dialysis, capacitive deionization, microwave hydrothermal, extraction by ionic liquids, thermal desalination processes, and direct contact membrane distillation methods [8,9,10]. Among these methods, however, no standards are provided due to restrictions on boron removal. These methods have various limitations, especially in low boron concentrations. For example, the reverse osmosis method is insufficient for removal of boron from seawater under normal pH conditions. Membrane technologies have begun to replace these techniques in recent years.

Membrane technology applications attract increasing attention of researchers due to their properties such as less chemical use, higher thermal and mechanical properties, better permeability and selectivity [11,12]. Membrane technology has greatly improved our capabilities in restructuring manufacturing processes and introducing new technologies for sustainable growth and energy [13,14]. In recent years, membrane based processes have taken considerable attention in separation science [15]. Polymer inclusion membranes (PIMs) are a class of liquid membranes used on a large-scale as chemical sensors. These membranes are easy to prepare and very stable [12,16,17,18,19].

The prepared membrane is used to separate the donor and acceptor aqueous phases, but does not contain an organic solvent to facilitate the transport of ions or molecules through PIMs. The chemical and physical properties of PIMs offer the opportunity to be specially designed and prepared for specific applications [11,20,21]. Two things are required in the preparation of membranes: the physical and covalent immobilization of the components in the polymer matrix. The separation process in PIM is a very attractive alternative to the removal and recovery of metal ions from wastewater, with a very high efficiency. The mechanical properties of PIMs are quite similar to the mechanical properties of filtration membranes, thus PIM-based systems exhibit many advantages, such as ease of use, minimal use of harmful chemicals, flexibility in membrane composition, desired selectivity, and separation efficiency [11,13,22,23]. For this purpose, in this study, new generation polymer membranes have been used for the recovery of boron from geothermal waters.

The maximum permissible limit of boron is given in the country for water samples as drinking, irrigation, and waste. The limits of boron concentration range from 0.5 mg/L to 5 mg/L. The recommended concentration of the World Health Organization (WHO) and the European Union (EU) are 2.4 mg/L (in 2011) and 1.0 mg/L in drinking water, respectively [24,25,26]. There are many methods as molecular spectrophotometric (UV–Vis), atomic spectrometric methods with mass/absorption spectrometry (MS, AAS), optical emission spectrometry (OES), and electroanalytical and chromatographic methods to detect and determine boron concentration. In particularly, the United States Environmental Protection Agency (EPA) recommends inductively coupled plasma mass spectrometry (ICP–MS) and optical emission spectrometry (OES) for water samples [26]. Total dissolved solid (TDS) is lower than the 0.2% recommended by the Environmental Protection Agency (EPA) for optimum determination of boron with ICP–MS and OES [27,28,29].

## 2. Materials and Methods

### 2.1. Reagents and Apparatus

Dichloromethane (CH_2_Cl_2_) (Sigma-Aldrich, Lenexa, KS, USA), H_2_SO_4_ (98% purity, Sigma-Merck, Germany), cellulose triacetate (MA = 72,000–74,000), 2-Nitrophenyl Octyl Ether (2-NPOE) (Honeywell Fluka, Wabash, IN, USA), ammonium acetate (NH_4_CH_3_COO), acetic acid (CH_3_COOH), hydrochloric acid (HCl), nitric acid (HNO_3_), and boric acid (H_3_BO_3_) (Merck, Germany) were used for pre-treatment and model working. Furthermore, sodium hydroxy (NaOH), sodium chloride (NaCl), and sodium sulfate (Na_2_SO_4_) were purchased from Sigma-Aldrich (USA). Ultra-pure water (resistivity 18.2 MΩ/cm) was used for all aqueous solutions and experimental steps. 

This study was carried out by using different instruments; a thermostatic (Poly Science 912, Niles, IL, USA) apparatus to keep the temperature controlled, magnetic stirrer (Yellow line MST basic, Germany) and reverse osmosis system (Human Corporation, Seoul, Korea) to water distillation. Moreover, weighing and determination of pH values of all samples were performed by Precisa XB 220A (Switzerland) brand analytical balance and WTW 720 brand pH meter, respectively. 

### 2.2. ICP–OES Analysis

ICP–OES of Perkin Elmer optima model 2100 DV was used for determination of boron. Working conditions of ICP–OES spectrometer were given in Table 1 below.

## 3. Results and Discussion

### 3.1. Effect of Boron Concentration

Membrane capacity is important in the transport process. The components that consist of the membrane structure have relative amounts. Current studies indicate that the membrane composition has a significant effect on the transport rates of target analytes [17,30,31]. However, research on this issue remains highly dispersed. The optimum relative quantities of the membrane components are only possible when the target analyte has a certain carrying capacity. The maximum transported amount of target analyte is defined as the membrane capacity. For this purpose, the experiments with different boron concentrations were carried out to depict the maximum amount of analyte that the membrane can transport. Because any change in membrane composition, or change in permeation flux, is directly related to membrane capacity. Four different boron concentrations (5, 10, 20, and 30 ppm) in 0.1 M HCl were used in experimental studies as a donor phase. The other experimental conditions were 0.1 M NaCl/NaOH as an acceptor phase and 1.75 mL 2-NPOE/1 g CTA/1.75 mL Aliquat 336 as a membrane phase at constant temperature (298K).

As seen in Table 2, excessive concentration of boron in the donor phase limited the transport of boron from a certain amount (20 ppm). This phenomenon is attributed to membrane capacity. The carrier in the membrane composition has a limited ability to form a complex with the target analyte. Constant rate, k, calculated according to the first order of the kinetic equation and the slopes of the curves in Figure 1, present k. An excessive amount of boron limited the kinetic values, so 20 ppm boron concentration was selected as an optimum value. P, J, and D were the permeability coefficient, flux, and diffusion coefficient of the transport process, respectively, calculated according to our previous studies [18,32,33,34]. The recovery factor (RF%) describes the efficiency of boron.

### 3.2. Effect of Donor Phase Concentration and Solvent Type

In PIM, the transport occurs in three steps. In the first step, the target analyte reacts with the carrier to form a complex at the source solution/membrane phase and replaces another molecule of the carrier after diffusing from a stagnant layer. In the second step, this complex, which occurs through the membrane phase, diffuses to a receiver phase. In the last step, the complex dissociate in the membrane/receiver phase solution interface and, thus, the target analyte stays in the receiver phase solution, which is the reverse of the process that occurs at the feed solution/membrane interface. Different solvents, which are given in Table 3, were used in the donor phase to reveal the effects of different mediums in the transport efficiency. The mediums are neutral, or close to the neutral region, except HCl. Although one the main driving forces is concentration gradient, the pH difference between the donor and acceptor phases is another important driving force in the transport process. Therefore, based on this definition, the boron transport had the highest value when 0.1 M HCl was used in the donor phase in the light of kinetic data (Table 3). The pH values of other mediums are close to each other, resulting in similar transport efficiency. Furthermore, high results obtained from the donor phase in the HCl medium, using 0.1 M HCl, which has the lowest pH value, gives very high kinetic results when compared to other HCl concentrations in Figure 2, drawn according to P and J data, obtained by using different HCl concentrations.

### 3.3. Effect of Acceptor Phase Solvent Type and Concentration

In PIM studies, interface transport mechanisms are observed at the membrane–aqueous interface, and are closely related with chemistry of the aqueous phase (both donor and acceptor phases) [12]. Potential gradient is one of the driving forces of the coupled transport ion for the transport that occurs through the membrane. In a typical PIM system, the target analyte transports with this ion to maintain the electro neutrality. This is a known counter or co-transport, depending on the transport direction of the coupled transport ion with the target analyte [11]. Here, in such a case, the potential gradient of protons can be arranged with pH solution, which is the driving force of the uphill transport of metal cations through the membrane. However, it should be noted that the distribution rate of the target analyte between the membrane phase and aqueous phase is associated with pH solution. In PIM studies, it is well known that the distribution rates of acidic and chelate extraction are highly dependent on pH, although Kp values are not usually reported at different pH values. 

According to the previous section, 0.1 M HCl concentration is an optimum pH value in the donor phase. However, the interface transport mechanisms are closely related (both the donor and acceptor phase), so the experiments are achieved with four different acceptor mediums too. Table 4 presents the obtained kinetic results for different acceptor mediums. As a result of the experiment with the use of NaOH in the acceptor phase, the recovery value (%) of boron is the highest with 60.46. To demonstrate the effect of different NaOH concentrations after achieving the highest recovery value, the experiments repeated with three different concentrations; the results are presented in Table 5. It can be seen that a better transport rate is achieved by using 0.1 M NaOH as the acceptor phase when compared to other concentrations.

### 3.4. Transport Mechanism

Essentially, the transport process in PIM involves the exchange of ionic species between the donor and acceptor phases, which are separated with the help of the membrane phase. Uphill transport of the target analyte through the membrane can be achieved by an appropriate ionic structure in the donor and acceptor phases. In other words, uphill transport is only achieved by the total analytic concentration of solute, while the downhill transport is related with chemical species, which are diffused through the membrane. In uphill transport, another driving force is the potential gradient of coupled transport ions through the membrane. Substantially, uphill and downhill transports are integral parts of a complex interface transport mechanism. However, in practice, these two driving forces are indistinguishable from each other [11]. This is due to the complexity of speciation that takes place on both sides of the membrane and within the membrane itself. One emphasizes the difference in the distribution ratio (Kp), while the other highlights the potential gradient of the coupled transport ion through the membrane.

H_3_BO_3_, which is used as the target analyte boron source in the donor phase, is partially ionized. H_2_BO_3_^−^ anions formed as a result of this partial ionization migrate through the donor phase/membrane phase diffusion layer (Figure 3). While Cl^−^ anions that are formed as a result of the interaction between the cationic carrier Aliquat 336 and H_3_BO_3_/H_2_BO_3_^−^ pairs in the membrane phase diffuse through the donor phase; the Aliquat 336–H_3_BO_3_/H_2_BO_3_^−^ ion pair, which formed in the membrane phase diffuses through the acceptor phase. We suggest that, Aliquat 336–H_3_BO_3_/H_2_BO_3_^−^ ion pairs in the acceptor/membrane diffusion layer dissociates with the effect of the acceptor phase pH difference. As H_3_BO_3_/H_2_BO_3_^−^ migrates to the acceptor phase, it replaces the Cl^−^ ions in the acceptor phase.

### 3.5. Application of Membrane System in Geothermal Waters and Selectivity

A geothermal water sample was applied to have an opinion in ability of the membrane selectivity, separation efficiency, and transport performance of the membrane system in real samples. A geothermal water sample was achieved from local geothermal supply. Geothermal waters contain silica, arsenic, and boron compounds at high concentrations. Boron is found in geothermal water as boric acid (H_3_BO_3_). Its concentration is higher in sedimentary rocks enriched with organic substances. Table 6 presents the cations in geothermal water that determinate with ICP–OES. A geothermal water sample was used as a donor phase in the transport process without changing other optimum experimental conditions.

In the presence of other metal cations in the geothermal water sample solution, boron was transported, selectively, with 59.85% recovery, when the other metal cations of RF values were evaluated. In the optimum transport conditions, the carrier was only selective towards boron among the other metal cations. This phenomenon arises as a big advantage in polymer membrane studies. Consequently, this study indicates the application of real samples in membrane studies, which are very practical and easy to use for large-scale applications. Table 7 presents the different techniques [35] with this study.

## 4. Conclusions

There are many various types of separation and purification methods for removal of boron. In this study, we improved a practical, easy, economic, recyclable, and selective transport process based on the polymeric membrane system. This study indicates the application of real samples in polymeric membrane systems. The chemical and physical properties of the prepared membrane were specially designed for selective and efficient transport of boron. The developed system based on polymeric membrane is a very attractive alternative to recovery and removal of boron from geothermal water, with selectivity and efficiency. Therefore, polymeric membrane based systems exhibit ease of use, basic, stable, desired selectivity and separation efficiency, flexibility in membrane composition, minimal use of harmful chemicals and inexpensive advantages in application of real samples. Thus, the developed system can be easily used in industrial scale applications, and it introduced new separation technology for sustainable removal of boron from geothermal water.

## Figures and Tables

**Figure 1 membranes-11-00291-f001:**
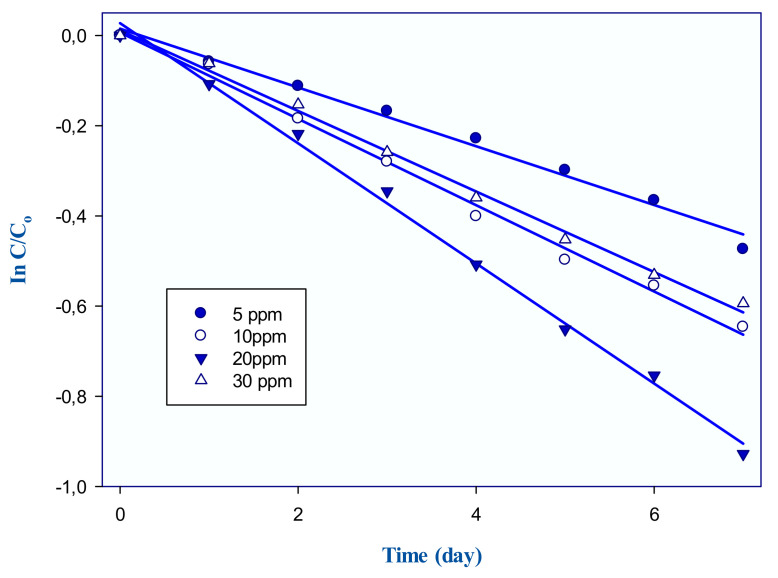
Constant rate of boron for each boron concentration.

**Figure 2 membranes-11-00291-f002:**
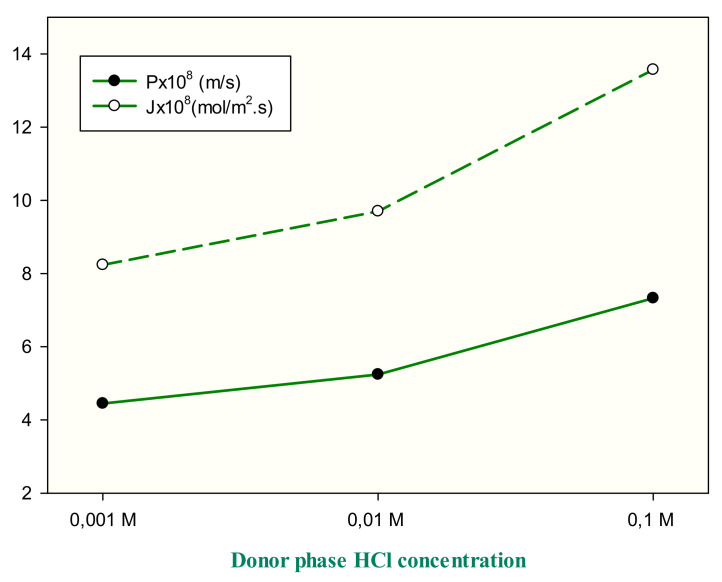
Effect of HCl concentration at the donor phase on the transport of boron.

**Figure 3 membranes-11-00291-f003:**
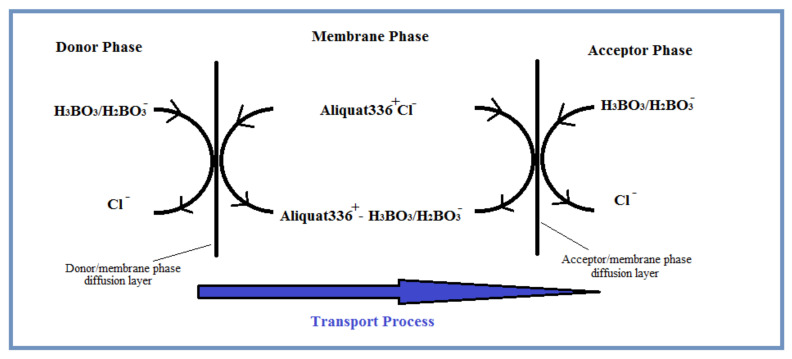
Suggested transport mechanism of boron.

**Table 1 membranes-11-00291-t001:** Sample introduction compartment characteristics and some inductively coupled plasma–optical emission spectrometry (ICP–OES) spectrometer parameters for boron determination.

Sample Introduction Compartment/Parameter	Type/Value
Torch	Perkin Elmer, Quartz Avio 500
Spray chamber	Perkin Elmer, Baffled Quartz Cyclonic Spray Chamber Avio 200
Nebulizer	Perkin Elmer, PFA-400 Micro Flow
Generator	1300 Watt
Plasma flow	16 L/min (Helium)
Plasma View	Axial
Auxiliary flow	0.2 mL/min
Nebulizer flow	0.65 mL/min
Sample flow rate	1.5 mL/min
Equilibration time	20 sec
Wavelength	249 nm
Interferences	-
Replicates	3

**Table 2 membranes-11-00291-t002:** Effect of boron concentration.

Boron Concentration (ppm)	k × 10^6^ (s^−1^)	P × 10^8^ (m/s)	J × 10^8^ (mol/m^2^.s)	D_0_ × 10^12^ (m^2^/s)	RF (%)
0.7540	0.7540	3.7367	6.9198	4.7551	37.72
1.1079	1.1079	5.4907	10.1679	5.5368	47.60
1.4785	1.4785	7.3273	13.5691	5.8174	60.46
1.0340	1.0340	5.1244	9.4896	5.4904	44.80
0.7540	0.7540	3.7367	6.9198	4.7551	37.72

Donor phase: four different boron concentrations (5, 10, 20, and 30 ppm) in 0.1 M HCl; acceptor phase: 0.1 M NaCl/NaOH; membrane phase: 1.75 mL 2-NPOE/1 g CTA/1.75 mL Aliquat 336.

**Table 3 membranes-11-00291-t003:** Effect of donor phase solvent type on the transport of boron.

Donor Phase Solvent Type	k × 10^6^ (s^−1^)	P × 10^8^ (m/s)	J × 10^8^ (mol/m^2^.s)	D_0_ × 10^12^ (m^2^/s)	RF (%)
0.1 M NaCl	0.5904	2.9260	5.4185	4.6048	30.50
0.1 M HCl	1.4785	7.3273	13.5691	5.8174	60.46
0.1 M Na_2_SO_4_	0.5581	2,7659	5.1220	4.5717	29.04
DDW	0.5228	2.5909	4.7981	4.4180	28.15

Donor phase: 20 ppm boron concentrations in four different solvent type; acceptor phase: 0.1 M NaCl/NaOH; membrane phase: 1.75 mL 2-NPOE/1 g CTA/1.75 mL Aliquat 336.

**Table 4 membranes-11-00291-t004:** Effect of acceptor phase solvent type on the transport of boron.

Acceptor Phase Solvent Type	k × 10^6^ (s^−1^)	P × 10^8^ (m/s)	J × 10^8^ (mol/m^2^.s)	D_0_ × 10^12^ (m^2^/s)	RF (%)
0.1 M Na_2_SO_4_	0.3583	1.7757	3.2883	4.4392	19.20
0.1 M NaCl	0.4369	2.1652	4.0096	4.7369	21.94
0.1 M NaOH	1.4785	7.3273	13.5691	5.8174	60.46
DDW	0.3247	1.6092	2.9799	4.2557	18.15

Donor phase: 20 ppm boron concentrations in 0.1 M HCl; acceptor phase: four different solvent type; membrane phase: 1.75 mL 2-NPOE/1 g CTA/1.75 mL Aliquat 336.

**Table 5 membranes-11-00291-t005:** Effect of NaOH concentration at acceptor phase on the transport of boron.

NaOH Concentration at Acceptor Phase	k × 10^6^ (s^−1^)	P × 10^8^ (m/s)	J × 10^8^ (mol/m^2^.s)	D_0_ × 10^12^ (m^2^/s)	RF (%)
0.01 M	0.8322	4.1243	7.6375	5.2371	37.80
0.05 M	1.0067	4.9892	9.2392	5.2517	45.60
0.1 M	1.4785	7.3273	13.5691	5.8174	60.46

Donor phase: 20 ppm boron concentrations in 0.1 M HCl; acceptor phase: 0.1–0.01 M NaOH/0.1 M NaCl; membrane phase: 1.75 mL 2-NPOE/1 g CTA/1.75 mL Aliquat 336.

**Table 6 membranes-11-00291-t006:** Concentration of detected cations in a geothermal water sample and RF values of the transport.

Cations	Initial Concentration (mg/L)
B	42.86
Na^+^	1294.254
K^+^	1463.926
Li^+^	17.56
Rb^+^	0.94
Cs^+^	1.83
Ca^+2^	37.51
Mg^+2^	0.08
Al^+3^	<0.001
Fe^+2^, Fe^+3^	0.07
Mn^+2^	0.01

**Table 7 membranes-11-00291-t007:** Different techniques for separation of boron.

Transport Type	Source of Boron	Initial Boron Concentration (mg/L)	Removal Efficiency (%)	Reference
Reverse osmosis	Seawater	5.1	>98	[36]
Electrodialysis	Saline solution	50	-	[37]
Forward osmosis	Model seawater	-	80	[38]
Membrane distillation (polyvinylidene fluoride membrane)	Seawater	5.37	90.50	[39]
Microfiltration process	Seawater	5.083	-	[40]
in this study	Geothermal water	42.86	59.85	

## Data Availability

The data presented in this study are openly available in results and discussion section of this study.

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
