# Peer review of "Separation of Boron from Geothermal Waters with Membrane System"

_membranes, 2021, doi:10.3390/membranes11040291_

Round 1

Reviewer 1 Report

The authors have described a very important study on removing boron from geothermal sources and this is indeed a topic of significant interest.  The use of a liquid membrane system, specifically a polymer inclusion membrane is also interesting. 

1.The data are presented clearly but no possible explanations or mechanisms have been provided. Without these, the paper only boils down to a "trial-and-error" study without any major scientific insight. In my view, this is the biggest flaw of this paper. 

For example, the authors have identified some optimum conditions for the donor phase and acceptor phase without ever trying to analyze why such conditions work well. Moreover, the results from an actual geothermal water are very important and it is interesting to note that competitive adsorption did not affect the boron uptake significantly. But once again, the authors did not provide any explanation or mechanistic insight into this. 

2.Lastly, the authors should provide the relevant references to support some of their statements - please see the attached pdf. 

Author Response

Thank you for your valuable comment and suggestions of our manuscript. We have revised the title according to your comment and modified the manuscript accordingly, and detailed corrections are listed  in the attachement file point by point in red:

Reviewer 2 Report

The manuscript describes a GOOD work AND is well presented. Authors need following points to be included before reconsideration.

1. Abstract should contain some quantitative information also.

2. English must be improved.

3. Novelty of the work be established.

  1. Results reported be compared in a tabular form to establish the superiority of the work.

  1. Authors need to add future prospective of the presented research in the conclusion part of the manuscript.

  2. Authors must need to incorporate following recent references related to water purification in the introduction part of the manuscript to make it more interesting for the readers.

  • Eng. Chem. Res.2011, 50, 10, 6325–6330
  • Eng. Chem. Prod. Res. Dev.1964, 3, 4, 304–306
  • ACS Omega2020, 5, 18, 10301–10314
  • ACS Earth Space Chem.2020, 4, 8, 1269–1280
  • https://doi.org/10.1080/19443994.2016.1176962
  • https://doi.org/10.1002/adsu.202070009
  • Environmental Science: Water Research & Technology 6 (11), 3080-3090
  • RSC advances 9 (69), 40565-40576
  • Cellulose 25 (3), 1961-1973

Author Response

(The authors gave the same response as above.)

Round 2

Reviewer 1 Report

Thank you for addressing the comments/questions.